# Thymic stromal lymphopoietin limits primary and recall CD8[+] T-cell anti-viral responses

Risa Ebina-Shibuya[1†‡], Erin E West[1†§], Rosanne Spolski[1], Peng Li[1], Jangsuk Oh[1], Majid Kazemian[1#], Daniel Gromer[1¶], Phillip Swanson[2**], Ning Du[1], Dorian B McGavern[2], Warren J Leonard[1*]

[1]Laboratory of Molecular Immunology, Immunology Center, National Heart, Lung, and Blood Institute National Institutes of Health, Bethesda, United States; [2]Viral Immunology & Intravital Imaging Section, National Institute of Neurological Disorders and Stroke, National Institutes of Health, Bethesda, United States

*For correspondence: wjl@helix.nih.gov

[†]These authors contributed equally to this work

Present address: [‡]Department of Respiratory Medicine, Tohoku University, Graduate School of Medicine, Sendai, Japan; [§]Laboratory for Complement and Inflammation Research, Immunology Center, NHLBI, NIH, Bethesda, United States; [#]Departments of Biochemistry and Computer Science, Purdue University, West Lafayette, United States; [¶]Department of Internal Medicine, Massachusetts General Hospital, Boston, United States; [**]Vaccine Immunology Program, Vaccine Research Center, NIAID/NIH, Gaithersburg, United States

Competing interests: The authors declare that no competing interests exist.

**Abstract** Thymic stromal lymphopoietin (TSLP) is a cytokine that acts directly on CD4[+] T cells and dendritic cells to promote progression of asthma, atopic dermatitis, and allergic inflammation. However, a direct role for TSLP in CD8[+] T-cell primary responses remains controversial and its role in memory CD8[+] T cell responses to secondary viral infection is unknown. Here, we investigate the role of TSLP in both primary and recall responses in mice using two different viral systems. Interestingly, TSLP limited the primary CD8[+] T-cell response to influenza but did not affect T cell function nor significantly alter the number of memory CD8[+] T cells generated after influenza infection. However, TSLP inhibited memory CD8[+] T-cell responses to secondary viral infection with influenza or acute systemic LCMV infection. These data reveal a previously unappreciated role for TSLP on recall CD8[+] T-cell responses in response to viral infection, findings with potential translational implications.

## Introduction

Influenza virus infection accounts for significant morbidity and mortality (*Davlin, 2016*; *Budd, 2016*), and understanding factors controlling the immune response to influenza is important for developing strategies for enhancing immunity and designing new therapies and vaccines. The absence of CD8[+] T cells delays influenza clearance (*Moskophidis and Kioussis, 1998*), demonstrating the importance of these cells in the control of infection by this virus. The main cellular targets of influenza are pulmonary epithelial cells, which once infected produce multiple inflammatory mediators that can alter the immune response to influenza infection. One of these mediators is TSLP, a pleiotropic cytokine with a range of actions, affecting cellular maturation, survival, and recruitment of cells. Although TSLP was initially reported to act on T cells indirectly through dendritic cells (*Soumelis et al., 2002*; *Ito et al., 2005*), it was later shown to also act directly on both mouse and human CD4[+] and CD8[+] T cells (*Rochman et al., 2007*; *Al-Shami et al., 2005*; *Shane and Klonowski, 2014*; *Rochman and Leonard, 2008*). TSLP has additional effects on B cells, neutrophils, mast cells, and eosinophils (*West et al., 2016*; *Rochman et al., 2009*; *Corren and Ziegler, 2019*). TSLP signals via a receptor comprising a TSLP-specific binding protein, TSLPR, and the IL-7 receptor α chain, IL-7Rα (CD127), thereby activating JAK1, JAK2, and STAT5 (*Rochman et al., 2010*). TSLP is expressed at barrier surfaces, and has been extensively studied in the context of T helper 2 (T$_H$2) type responses and shown to promote the progression of T$_H$2-mediated diseases, including asthma, atopic dermatitis, and allergic inflammation (*Al-Shami et al., 2005*; *Divekar and Kita, 2015*; *Yoo et al., 2005*; *Zhou et al., 2005*), as well as immune responses to the intestinal pathogen, *Trichuris muris* (*Taylor et al., 2009*),

but the role of TSLP on CD8$^+$ T-cell responses is less-well characterized. There are conflicting reports of the role of TSLP on CD8$^+$ T cells during primary influenza infection (*Shane and Klonowski, 2014*; *Plumb et al., 2012*; *Yadava et al., 2013*), and the effects of TSLP on memory CD8$^+$ T cells and secondary responses to acute viral infections have not been characterized. Here, we used an adoptive co-transfer model of WT and TSLPR-deficient mice (the gene encoding TSLPR is the *Crlf2* gene, so these mice are designated as *Crlf2$^{-/-}$*) virus-specific CD8$^+$ T cells to analyze the direct actions of TSLP on CD8$^+$ T cells during both primary and secondary responses to influenza virus infection, as well as the role of this cytokine in naïve and memory CD8$^+$ T-cell homeostasis. We also assessed the role of TSLP in the context of an acute systemic infection caused by LCMV.

## Results

### TSLP acts directly on CD8$^+$ T cells during primary influenza infection

To assess the role of TSLP on CD8$^+$ T-cell responses during influenza infection, we adoptively transferred P14 T cells (TCR transgenic CD8$^+$ T cells specific for LCMV glycoprotein 33, gp33) into WT mice. We then infected these mice intranasally one day later with influenza strain PR8-33, which represents the PR8 strain genetically modified to express gp33 (*Mueller et al., 2010*), and then examined TSLPR expression over time in lungs and spleen (see schematic, upper part of *Figure 1A*). TSLPR was expressed on naïve (CD44$^{low}$) CD8$^+$ T cells, with high expression on virus-specific CD8$^+$ T cells in both the lungs and spleen by day six post-infection (*Figure 1A*), with a subsequent decrease evident at days 14 and 33 (*Figure 1A*), suggesting that TSLP might directly act on virus-specific CD8$^+$ T cells, and indeed increased *Tslp* mRNA expression has been observed during influenza infection (*Shane and Klonowski, 2014*; *Yadava et al., 2013*).

To determine whether there was a direct effect of TSLP on virus-specific CD8$^+$ T cells during influenza infection, we co-transferred equal numbers of congenically-labeled naïve WT and *Crlf2$^{-/-}$* P14 T cells into WT recipient mice, infected them intranasally with PR8-33, and assessed TSLPR expression as well as WT and *Crlf2$^{-/-}$* T-cell numbers and function, both at the peak of the response (day 8) and after the formation of memory cells (>day 30 p.i.) (see schematic in *Figure 1B*, upper panel and transferred cells lower panel). TSLPR was highly expressed on the virus-specific WT CD8$^+$ T cells but not *Crlf2$^{-/-}$* cells in all tissues assessed at day 8 p.i.: lungs, mediastinal (draining) lymph node, spleen, and bronchoalveolar lavage (BAL) fluid, with highest expression seen in the BAL fluid (*Figure 1C*). At day 8 p.i., there was a modest increase in *Crlf2$^{-/-}$* T cells compared to WT cells in the lungs, mediastinal lymph node, and spleen, but not in BAL fluid (*Figure 1D* and *Figure 1—figure supplement 1*); the results were qualitatively similar when the experiment was performed using Th1.1$^+$/Thy1.1$^+$ WT P14T cells and Thy1.1$^+$/Thy1.2$^+$ *Crlf2$^{-/-}$*cells (*Figure 1—figure supplement 1A*) or with Th1.1$^+$/Thy1.2$^+$ WT P14T cells and Thy1.1$^+$/Thy1.1$^+$ *Crlf2$^{-/-}$* cells (*Figure 1—figure supplement 1B*), with composite results shown in (*Figure 1D*). Thus, the difference was due to WT versus *Crlf2$^{-/-}$* differences rather than differences that might exist between Thy1.1$^+$/Thy1.1$^+$ and Thy1.1$^+$/Thy1.2$^+$ genetic backgrounds. Some variation in the expression of CD44 (*Figure 1—figure supplement 2A*) and KLRG1 (*Figure 1—figure supplement 2B*) was observed, but the differences were modest, and there were similar levels of granzyme B (*Figure 1—figure supplement 2C*) and percentages of WT and *Crlf2$^{-/-}$* cells expressing IFN-γ, TNF-α, or both cytokines after ex-vivo stimulation with cognate peptide (GP33) (*Figure 1—figure supplement 2D*). Because TSLP signals via a dimer of the TSLP-specific receptor chain (TSLPR) and IL-7Rα, we considered the possibility that the absence of TSLPR leads to the compensatory induction of IL-7Rα (CD127), which might result in an IL-7-dependent increase in the number of *Crlf2$^{-/-}$* cells, but expression of IL-7Rα was similar in WT and *Crlf2$^{-/-}$* cells (*Figure 1E and F*). The increased cellularity in lungs, lymph node, and spleen (*Figure 1D*) indicated that TSLP negatively regulates CD8$^+$ T cell effector responses during acute pulmonary influenza infection; however, the proportions of WT and *Crlf2$^{-/-}$* memory CD8$^+$ T cells were not significantly different during the memory phase in lungs, lymph nodes, spleen, and BAL fluid (*Figure 1G and H*).

### TSLP affects homeostasis of naïve but not memory CD8$^+$ T cells

TSLP has been shown to play a direct role in the survival and homeostasis of naïve CD8$^+$ T cells both in vivo and in vitro, inducing enhanced BCL-2 expression and higher proliferation, with decreased apoptosis in naïve CD8$^+$ T cells in vitro and higher survival/homeostasis of these cells after cell

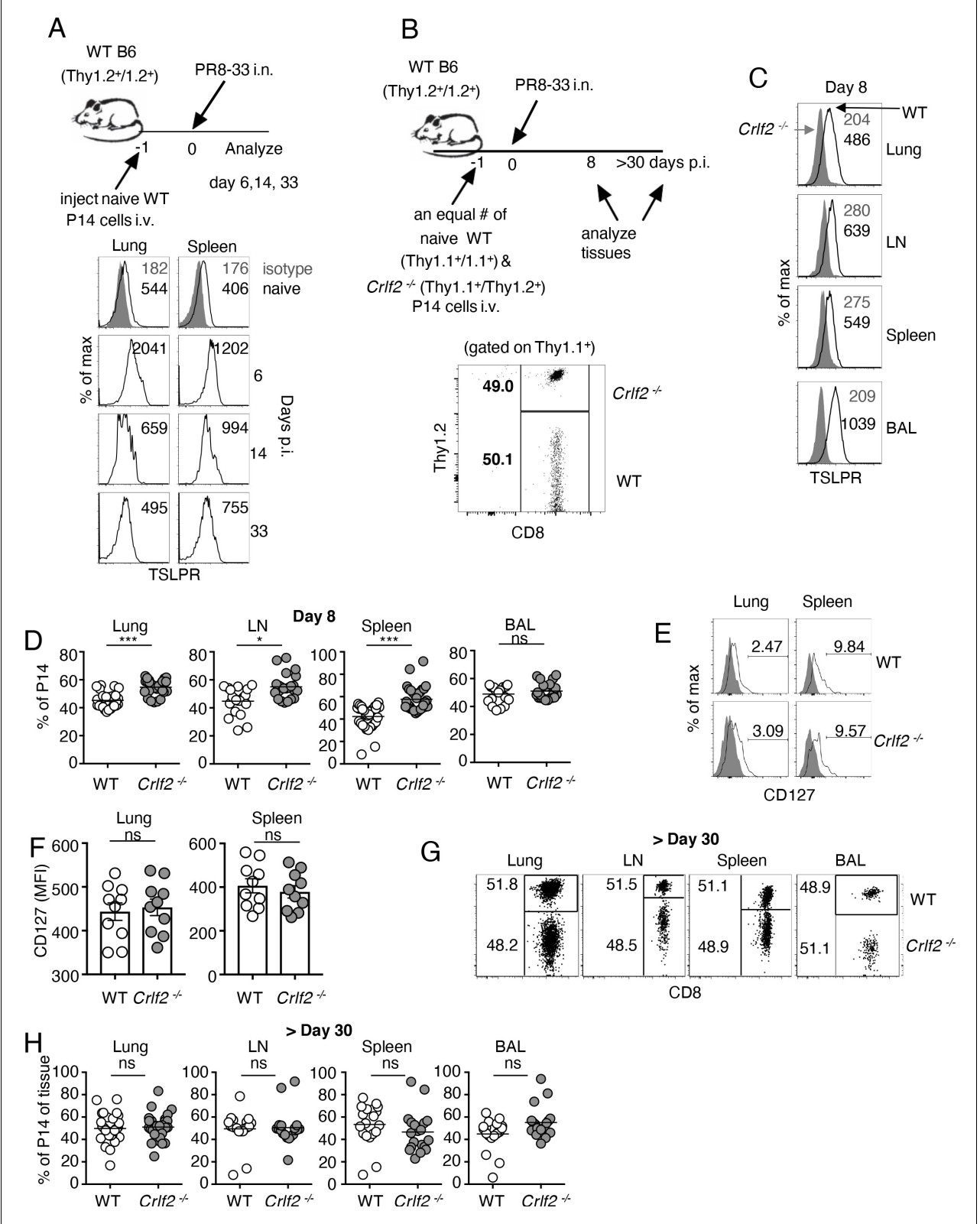

**Figure 1.** TSLP acts directly on CD8[+] T cells during primary influenza infection. (**A**) TSLPR expression on influenza-specific CD8[+] T cells (P14 tg) during primary influenza infection. Top panel, experimental design. Bottom panel, flow cytometric analysis. Naïve cells were gated on CD44[lo] cells. (**B–F**) (**B**) Top panel, experimental design for C-H, where 2.5 × 10[4] of WT (Thy1.1[+]/1.1[+]) and *Crlf2*[-/-] (Thy1.1[+]/1.2[+]) P14 T cells were co-transferred into naïve WT C57BL/6 mice (Thy1.2[+]/1.2[+]), except in one experiment the markers were reversed, with the WT cells Thy1.1[+]/1.2[+] and the *Crlf2*[-/-] P14 T cells were

*Figure 1 continued on next page*

*Figure 1 continued*

Thy1.1$^+$/Thy1.1$^+$ (see also *Figure 1—figure supplement 1* and **G**). Bottom panel, Similar numbers of WT P14 cells and *Crlf2$^{-/-}$* P14 cells were present. On the following day, the mice were infected intranasally (i.n.) with 10$^3$ EID$_{50}$ of PR8-33. Mice were analyzed at Day 8 p.i. (**C–F**) or at a memory time point (>day 30 p.i.) (**G and H**). (**C**) TSLPR expression on influenza-specific P14 CD8$^+$ T cells in the tissues on day 8 p.i. (**D**) Proportion of WT and *Crlf2$^{-/-}$* T cells at day 8 p.i. in the tissues (shown are combined data from three independent experiments). (**E and F**) The expression of CD127 on WT and *Crlf2$^{-/-}$* P14 cells in lungs and spleen. Shown are a representative flow cytometry plot (**E**) and summary of MFI data for CD127 expression (**F**). (n = 10). Data are mean ± SEM. (**G and H**) The proportion of WT and *Crlf2$^{-/-}$* P14 cells of transferred cells in BAL, lungs, LN, and spleen at a memory time point, shown as a representative flow cytometry plot (**G**) and combined data from three independent experiments (**H**). ns = not significant; *p<0.05; ***p<0.005, using a two-tailed paired students t-test. Data shown are representative of at least two independent experiments.

The online version of this article includes the following figure supplement(s) for figure 1:

**Figure supplement 1.** Thy1.1/Thy1.1 versus Thy1.1/Thy1.2 genetic background differences do not explain the different number of WT versus *Crlf2$^{-/-}$* P14 T cells after influenza infection.

**Figure supplement 2.** TSLP does not affect the transition to effector cells, cytokine secretion, or granzyme B levels during primary infection.

transfer into naïve hosts (*Rochman and Leonard, 2008*). We therefore investigated whether TSLP could enhance memory CD8$^+$ T-cell survival/homeostasis. We isolated P14 CD8$^+$ T cells from the spleens of naïve P14 mice or from the spleens of mice that were seeded with naïve P14 T cells and then infected with PR8-33 virus for more than 30 days to create memory P14 T cells. When naïve P14 T cells were cultured without anti-CD3 + anti-CD28, cell survival was >90% at 4 hr in the absence or presence of TSLP (*Figure 2A*, 1st and 2nd lanes). Survival was lower at 24 hr in medium, but TSLP significantly enhanced the survival (*Figure 2A*, 3rd and 4th lanes). However, when analogous cells were stimulated with anti-CD3 + anti-CD28, basal survival was higher and TSLP no longer further increased survival (*Figure 2A*, last two lanes). When we performed the same analysis on memory CD8$^+$ T cells, cell survival was also lower at 24 than at 4 hr, without a statistically significant effect of TSLP (*Figure 2B*). We next examined the role of TSLP in naïve and memory CD8$^+$ T-cell homeostasis in vivo. We co-transferred equal numbers of congenically marked WT and *Crlf2$^{-/-}$* naïve P14 cells into WT mice (*Figure 2C*) and found fewer *Crlf2$^{-/-}$* than WT cells at days 9–11 post-transfer (*Figure 2D*, left panels), consistent with a previous study that suggested that TSLP is important for naïve CD8$^+$ T-cell homeostasis (*Rochman and Leonard, 2008*). In contrast, both WT and *Crlf2$^{-/-}$* P14 memory CD8$^+$ T cells persisted similarly following transfer into naïve hosts (*Figure 2D*, right panels).

## TSLP affects gene expression in naïve and memory CD8$^+$ T cells in vitro

Given the differential impact of TSLP on naïve versus memory CD8$^+$ T cells, we sought to elucidate the mechanism and performed RNA-sequencing analysis on naïve and memory CD8$^+$ T cells stimulated in vitro with or without TSLP and in the absence (*Figure 3A* and *Supplementary file 1*) or presence (*Figure 3B* and *Supplementary file 2*) of TCR stimulation. The gating strategy for the memory CD8$^+$ T cells is shown in *Figure 3—figure supplement 1*. As expected, TSLP treatment of naïve cells and memory cells resulted in a number of genes up- or down-regulated in each treatment group. There were more down-regulated than up-regulated genes with TSLP except at the 4 hr time point without TCR activation, suggesting that TSLP was a mediator of gene repression. In naïve cells, the number of upregulated genes was lower at 24 than at 4 hr, independent of TCR stimulation, but in memory cells, the number of upregulated genes was higher at 24 than at 4 hr (*Figure 3A and B*), again indicating differences in the effect of this cytokine on naïve and memory T cells. We analyzed the genes in naïve versus memory CD8$^+$ T cells without TCR stimulation at 24 hr (overlap is shown in the Venn diagram in *Figure 3C*; list of genes in *Supplementary file 3*), and found that only a few genes were shared between naïve cells and memory cells, indicating distinctive effects of TSLP on these different cell types. Ten genes including *Alcam*, *Nfil3*, *Bcat1*, *Olfr613*, *Nr4a2*, *Bloc1s3*, *Zfp488*, *Cpox*, *Zfp457*, and *Ap1s3* were down-regulated in both naïve and memory CD8$^+$ T cells (*Figure 3C*, genes in blue in *Figure 3D*), whereas five genes including *Bcl2*, *Pole2*, *Socs3*, *Nek6*, and *Tfrc* were up-regulated in both cell types (*Figure 3C*; genes in red in *Figure 3D*). Up-regulation of *Bcl2* is consistent with higher cell viability after TSLP stimulation in vitro (*Figure 2*). Interestingly, seven genes were down-regulated in naïve CD8$^+$ T cells but up-regulated in memory CD8$^+$ T cells (*Hspa1b*, *Gadd45g*, *H2-ab1*, *Isg15*, *Socs2*, *Fcer1g*, and *Sfxn2*) (genes in black in *Figure 3D*), consistent with some potentially opposing/distinctive actions for TSLP in naïve versus memory CD8$^+$ T cells. Moreover, there were a number of genes significantly up- or down- regulated by TSLP in memory cells

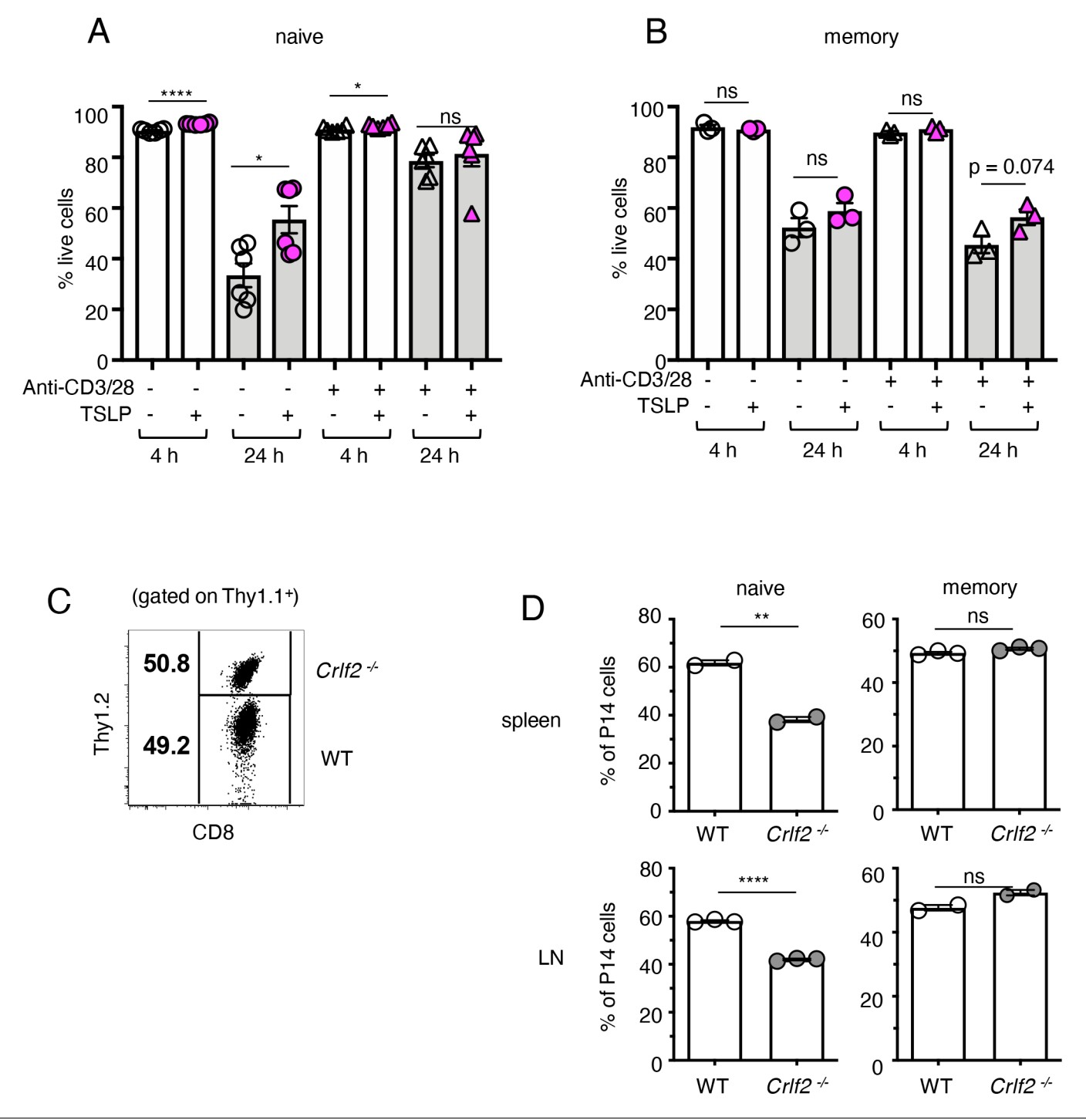

**Figure 2.** TSLP affects homeostasis of naïve but not memory CD8+ T cells. (**A and B**) (**A**) Naïve P14 CD8+ T cells were purified and plated with either media or TSLP with or without plate-bound anti-CD3 and soluble anti-CD28. Cell viability was assessed by gating on CD8+ T cells and live/dead staining after 4 and 24 hr. (n = 6). Data are mean ± SEM. (**B**) Memory P14 CD8+ T cells were purified and plated with either media or TSLP with or without soluble anti-CD3 and soluble anti-CD28. Cell viability was assessed by gating on CD8+ T cells and live/dead staining after 4 and 24 hr. (n = 3). Data are mean ± SEM. (**C**) Naïve P14 CD8+ T cells were purified and equal numbers of WT (Thy1.1+/1.1+) and *Crlf2-/-* (Thy1.1+/1.2+) P14 CD8+ T cells were co-transferred into naïve WT C57BL/6 mice (Thy1.2+/1.2+). Shown are representative flow cytometry plots (gated on live Thy1.1+ CD8+ cells) of the combined naïve WT and *Crlf2-/-* P14 cells pre-transfer. (**D**) Percent of naïve and memory WT and *Crlf2-/-* cells of total transferred P14 T cells on day 9–11 post cell transfer. (n = 2 or 3). Data are mean ± SEM. For A, B and D: Data are mean ± SEM. ns = not significant; *p<0.05; **p<0.01; ****p<0.0001 using a two-tailed paired students t-test. Data shown are representative of at least two independent experiments.

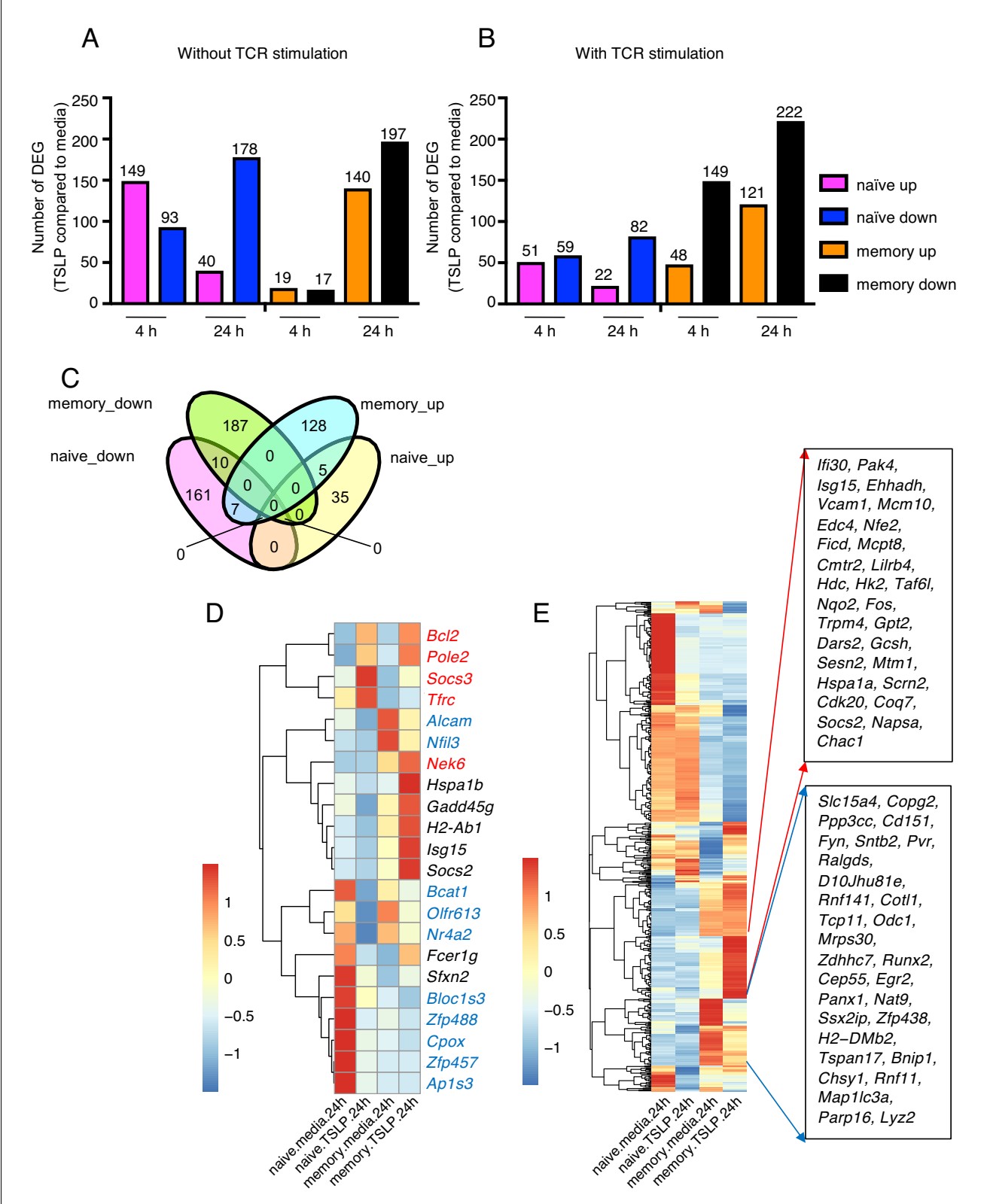

**Figure 3.** TSLP modulates gene expression on naïve and memory CD8[+] T cells in vitro. RNA-Seq performed on sorted naïve and memory CD8[+] T cells after 4 or 24 hr incubation with medium or TSLP with or without TCR stimulation. Shown are the number of differentially expressed genes (FC >1.5, FDR < 0.05). (**A and B**) Number of genes affected by TSLP in CD8[+] T cells not stimulated (**A**) or stimulated (**B**) with anti-CD3 + anti-CD28. (**C**) Venn diagram showing the number of genes whose expression was upregulated or downregulated by TSLP without TCR stimulation at 24 hr of incubation in

*Figure 3 continued on next page*

Figure 3 continued

naïve and memory CD8$^+$ T cells. (D) Genes regulated by TSLP both in naïve and memory CD8$^+$ T cells, color-coded as indicated in the text. (E) Genes differentially regulated by TSLP in naïve vs. memory cells. Highlighted are those that were selectively induced or repressed after stimulation with TSLP for 24 hr.

The online version of this article includes the following figure supplement(s) for figure 3:

**Figure supplement 1.** Gating strategy for sorting memory CD8$^+$ T cells for in vitro stimulation.

relative to naïve cells (*Figure 3E* and *Supplementary file 3*). One of the induced genes, *Hk2*, encodes a key rate-limiting enzyme in glycolysis, a process that is active in effector T cells and promotes T cell proliferation (*Robey and Hay, 2006*; *O'Neill et al., 2016*). Three of the genes whose expression was diminished by TSLP were *Runx2*, *Egr2*, and *Panx1*. RUNX2 is a transcription factor that promotes long-term persistence of antiviral CD8$^+$ memory T cells (*Olesin et al., 2018*), EGR2 is critical for normal differentiation of naïve T cells and for regulating antigen-specific immune responses to influenza viral infection (*Du et al., 2014*), and PANX1 was reported to influence memory T cell maintenance (*Wanhainen et al., 2019*), collectively indicating that there are distinctive effects of TSLP on naïve and memory T cells. The potential role(s) of these genes during influenza infection remains to be further elucidated.

## TSLP has a direct inhibitory effect on secondary CD8$^+$ T-cell responses during influenza infection

To further understand the biological consequences of TSLP on memory cells, we next co-transferred equal numbers of congenically marked WT and *Crlf2$^{-/-}$* memory P14 cells (>30 days after influenza infection) into naïve WT mice and infected these mice with PR8-33 intranasally the following day (schematic in *Figure 4A*, left; CD44 expression on WT and *Crlf2$^{-/-}$* memory P14 cells and ratio of transferred cells *Figure 4A*, right panels). On day 8 p.i., virus-specific secondary effector CD8$^+$ T cells in BAL fluid, lungs, lymph nodes, and spleen expressed high levels of TSLPR (*Figure 4B*). There tended to be higher expression than observed on primary effector CD8$^+$ T cells at day 8 p.i. (*Figure 1C*), but the highest TSLPR expression was still observed on BAL fluid cells (*Figure 4B*). When we assessed the secondary effector P14 responses on day 8 p.i., effector P14 T cells were present in all tissues, with a markedly increased proportion of *Crlf2$^{-/-}$* P14 T cells in lungs, lymph nodes, spleen, and BAL fluid (*Figure 4C and D*). These data demonstrate that TSLP constrains secondary CD8$^+$ T-cell responses during influenza infection. However, the percentage of cells expressing both IFNγ and TNFα was similar in WT and *Crlf2$^{-/-}$* mice after ex-vivo stimulation with cognate peptide (GP33), with only modest differences in the percentage of cells producing only IFNγ or TNFα alone (*Figure 4—figure supplement 1A–C*).

We next assessed the effects of TSLP signaling on CD8$^+$ T-cell recall responses at day 8 after secondary infection by RNA-Seq. The gating strategy for WT and *Crlf2$^{-/-}$* P14 cells is shown in *Figure 4—figure supplement 2*. Compared with WT cells, 20 genes were induced and nine genes were repressed in *Crlf2$^{-/-}$* cells (*Figure 4E* and *Supplementary file 4*); we confirmed that *Crlf2* gene expression was absent in *Crlf2$^{-/-}$* cells (*Figure 4F*). Two genes whose expression was markedly increased were *Eps8l1* (epidermal growth factor receptor pathway substrate 8-like 1) and *Eaf2*. *Eps8l1* has been reported to upregulate cell cycle genes, induce chemokines and enhance migration of some cancer cells (*Yang et al., 2019*; *Huang et al., 2018*; *Zeng et al., 2018*), whereas *Eaf2* acts as an upstream modulator of non-canonical Wnt signaling, and has been suggested to suppress oxidative stress–induced apoptosis through inhibition of caspase 3 production and activation of Wnt3a signaling (*Feng and Guo, 2018*). By enhancing migration and decreasing apoptosis the upregulation of these genes may help to explain the greater numbers of *Crlf2$^{-/-}$* cells we observed in the recall CD8$^+$ T-cell responses (*Figure 4C and D*). Several *Ifitm* (interferon induced transmembrane) family members were also more highly expressed in *Crlf2$^{-/-}$* cells (*Figure 4G* and *Supplementary file 4*). These proteins confer cellular resistance to many viruses in both mice and humans (*Zhao et al., 2018*; *Liao et al., 2019*; *Bailey et al., 2014*), and IFITM3 is known to contribute to the control of influenza A virus (*Brass et al., 2009*). Thus, the induction of *Ifitm* family genes might protect cells, limiting cell death after viral infection and leading to greater numbers of *Crlf2$^{-/-}$* cells after secondary

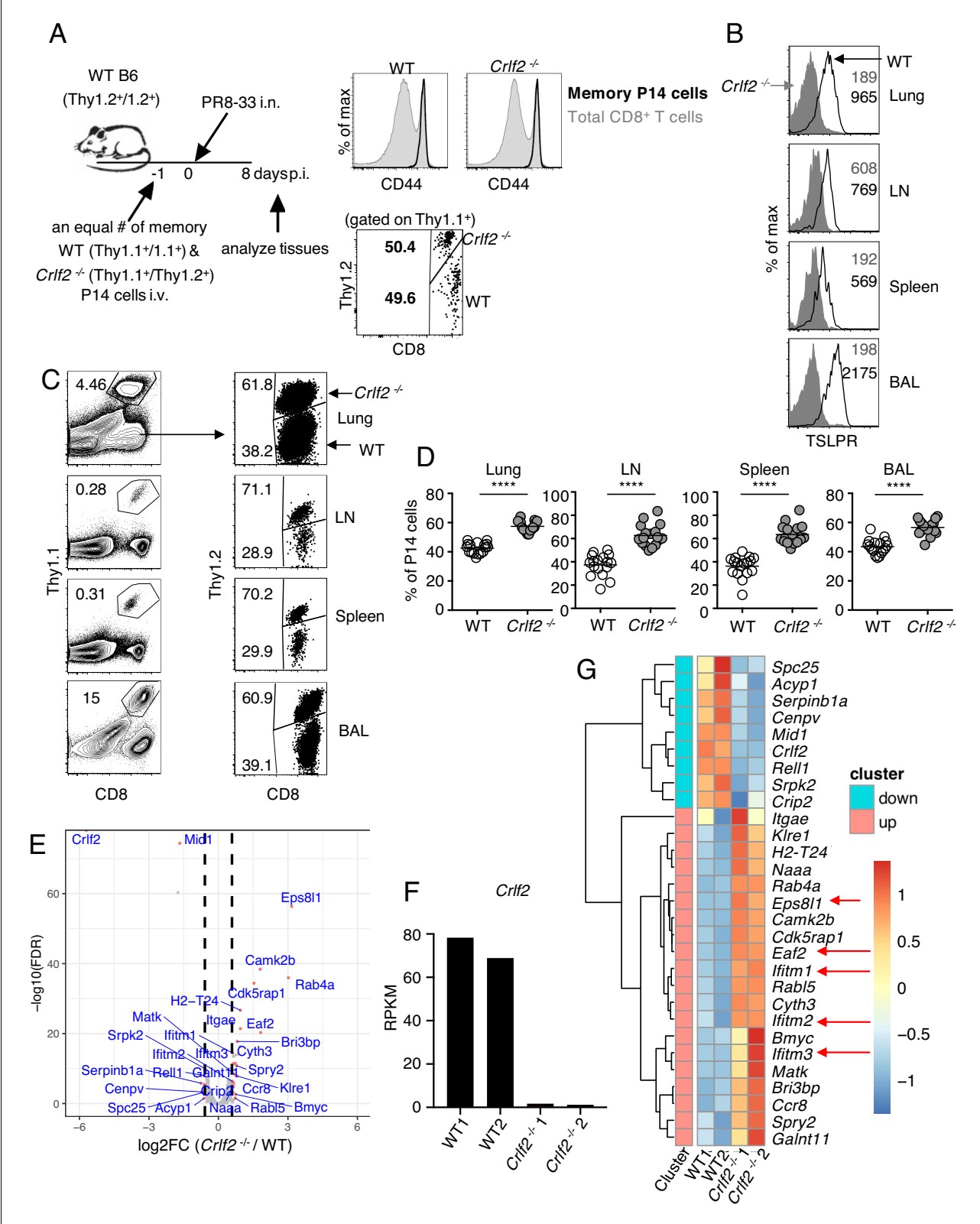

**Figure 4.** TSLP has a direct inhibitory effect on secondary CD8⁺ T-cell responses during influenza infection. (A) Left panel, experimental design for (B–G), where P14 T cells were isolated from P14 WT (Thy1.1⁺/1.1⁺) and P14 *Crlf2⁻/⁻* (Thy1.1⁺/1.2⁺) mice and then separately injected into naïve WT mice. Each mouse was then infected with influenza PR8-33 i.n. for >30 days, and WT P14 cells and *Crlf2⁻/⁻* P14 cells were separately isolated. These cells were all CD44^hi (right upper panel); 10⁴ cells from each type of mouse were then co-transferred into recipient naïve WT C57BL/6 mice (Thy1.2⁺/1.2⁺) on day
*Figure 4 continued on next page*

*Figure 4 continued*

−1, and similar numbers of WT P14 cells and *Crlf2*⁻/⁻ P14 cells were present (right lower panel). On the following day (day 0), each mouse was infected with $10^3$ EID$_{50}$ of PR8-33 i.n. (**B–D**) Mice were analyzed at day 8 p.i. (**B**) The expression of TSLPR on CD8⁺ T cells in BAL fluid, lungs, lymph node, and spleen. (**C**) Representative flow cytometry plots showing the percent of transferred P14 cells (gated on live lymphocytes) and proportion of WT and *Crlf2*⁻/⁻ cells within the transferred population at day 8 p.i. with influenza in the tissues. (**D**) Percentage of P14 WT and *Crlf2*⁻/⁻ T cells at day 8 p.i. with influenza in BAL fluid, lungs, LN, and spleen (combined data from three independent experiments are shown). (**E–G**) RNA-Seq was performed on cells from WT versus *Crlf2*⁻/⁻ mice. (**E**) Differentially expressed genes are shown. The dashed lines correspond to $\log_2(1.5)=0.585$. (**F**) Expression of *Crlf2* in cells from WT and *Crlf2*⁻/⁻ mice. (**G**) Heatmap of differentially expressed genes from WT or *Crlf2*⁻/⁻ CD8⁺T cells at day 8 p.i. with secondary influenza infection. Shown is the scale for fold induction or repression. Data are mean ± SEM. ns = not significant; ****p<0.0001 using a two-tailed paired students t-test. Data shown are representative of at least two independent experiments.

The online version of this article includes the following figure supplement(s) for figure 4:

**Figure supplement 1.** IFN-γ and TNF-α expression in WT versus *Crlf2*⁻/⁻ mice during secondary infection.
**Figure supplement 2.** Gating strategy for sorting WT and *Crlf2*⁻/⁻CD8⁺ T cells for RNA-sequencing.

infection, helping to explain the negative effect of TSLP on the expansion of CD8⁺ T cells in response to influenza virus infection.

## TSLP affects virus-specific CD8⁺ T cell responses during primary and recall LCMV infection

We next investigated whether the effects of TSLP on primary CD8⁺ T-cell responses that we observed with influenza (where infection is at a barrier surface where TSLP is expressed) might extend to a systemic infection such as acute LCMV infection. Using the same adoptive co-transfer method of WT and *Crlf2*⁻/⁻ P14 cells (**Figure 5A**) that we used for influenza infection (**Figure 4A**), we found that virus-specific P14 CD8⁺ T cells expressed more TSLPR at day 8 after LCMV infection in all tissues analyzed (blood, lungs, lymph node, and spleen) (**Figure 5B**), analogous to what we observed after influenza infection in lungs, lymph node, spleen, and BAL fluid (**Figure 1C**). Moreover, TSLP protein was also increased in the lungs at day 8 after LCMV infection (**Figure 5—figure supplement 1**). At day 8 p.i. with LCMV, WT, and *Crlf2*⁻/⁻ P14 T cell numbers were similar in most tissues examined, but *Crlf2*⁻/⁻ cells were modestly more abundant in the spleen (**Figure 5C**). At a memory time point, CD8⁺ T cells also expressed TSLPR (**Figure 5D**), with more *Crlf2*⁻/⁻ than WT P14 cells in the blood and lymph nodes of mice (**Figure 5E**), indicating that TSLP influences CD8⁺ memory T cell numbers in at least some tissues after acute LCMV infection.

To determine if TSLP also limited the secondary response to an acute systemic infection, analogous to what we observed with influenza infection, we next adoptively transferred equal numbers of LCMV memory WT and *Crlf2*⁻/⁻ P14 cells into naïve mice and infected these mice with LCMV Armstrong intraperitoneally (**Figure 5F**). At day 8 p.i., there were more *Crlf2*⁻/⁻ secondary effector P14 T cells than corresponding WT effector P14 T cells in blood, lungs, inguinal lymph nodes, and spleen, with *Crlf2*⁻/⁻ P14 T cells making up greater than 60% of the P14 population (**Figure 5G**). Thus, TSLP negatively regulated the secondary CD8⁺ T cell response to either an acute local lung infection (influenza) or an acute systemic infection (LCMV), revealing a previously unappreciated role for TSLP in directly modulating memory CD8⁺ T cell recall responses.

## Discussion

TSLP has been studied extensively in the context of T$_H$2-type immunity, and we previously demonstrated that TSLP can act directly on CD4⁺ and CD8⁺ T cells, but its roles in CD8⁺ T cell responses during viral infection remain poorly understood. Although TSLP is induced by viral infection of CD8⁺ T cells, there have been conflicting reports regarding its actions on CD8⁺ T cells during the primary response to influenza infection. One study using *Crlf2*⁻/⁻ mice indicated that TSLP does not affect the control of influenza infection nor affect virus-specific CD8⁺ T cell responses during primary infection (**Plumb et al., 2012**). Another report concluded that TSLP enhances CD8⁺ T-cell responses during primary influenza infection, but that this was not due to a direct action on CD8⁺ T cells and instead was an indirect effect on CD8⁺ T-cell responses resulting from TSLP-induced IL-15 production by dendritic cells (**Yadava et al., 2013**). Finally, a third study used an adoptive co-transfer model of WT and *Crlf2*⁻/⁻ TCR transgenic ovalbumin-specific CD8⁺ T cells (OT-I cells) and found that

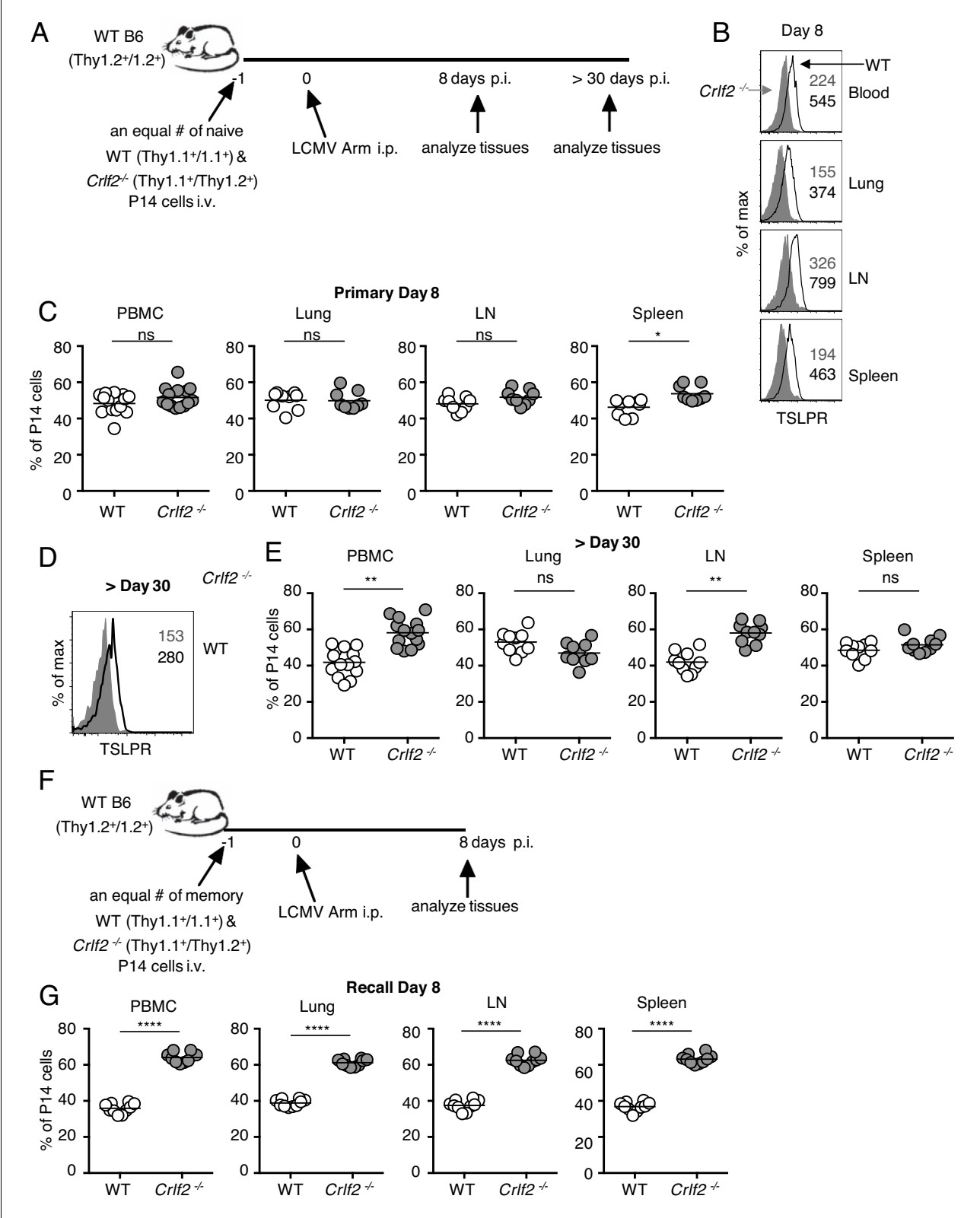

**Figure 5.** Direct actions of TSLP on virus-specific CD8[+] T cells during primary and recall LCMV infection. (A) Schematic of LCMV infection experiment. $10^3$ of WT (Thy1.1[+]/1.1[+]) and $Crlf2^{-/-}$ (Thy1.1[+]/1.2[+]) P14 naïve T cells were co-transferred into naïve WT C57BL/6 mice (Thy1.2[+]/1.2[+]) and on the following day the mice were infected i.p. with $2 \times 10^6$ pfu of LCMV Armstrong. Mice were analyzed at day 8 p.i. (B and C) or at a memory time point (>day 30 p.i.) (D and E). (B) TSLPR expression was assessed on LCMV-specific P14 CD8[+] T cells in the tissues on day 8 p.i. (C) Proportion of WT and

*Figure 5 continued on next page*

*Figure 5 continued*

*Crlf2⁻/⁻* T cells at day 8 p.i. in the tissues (combined data from two independent experiments shown). (**D**) TSLPR expression on memory P14 T cells from a mouse seeded with P14 T cells and infected with LCMV Armstrong i.p. for >30 days. (**E**) The proportion of WT and *Crlf2⁻/⁻* P14 cells of transferred cells in the tissues at >30 days, a memory time point (combined data from two independent experiments shown). (**F**) Schematic of LCMV infection experiment. WT (Thy1.1⁺/1.1⁺) and *Crlf2⁻/⁻* (Thy1.1⁺/1.2⁺) P14 T cells were isolated from mice seeded with P14 T cells and infected with LCMV Armstrong i.p. for >30 days, and equal numbers of WT and *Crlf2⁻/⁻* cells ($10^3$ of each population) were co-transferred into naïve WT C57BL/6 mice (Thy1.2⁺/1.2⁺) on day −1. On the following day (day 0), the mice were infected with $2 \times 10^6$ pfu LCMV Armstrong i.p. Mice were analyzed at day 8 p.i. (**G**) Proportion of WT and *Crlf2⁻/⁻* T cells at day 8 p.i. with LCMV in the tissues, (combined data from two independent experiments shown). Data are mean ± SEM. ns = not significant; *p<0.05; **p<0.01; ****p<0.0001 using a two-tailed paired students t-test. Data shown are representative of at least two independent experiments.

The online version of this article includes the following figure supplement(s) for figure 5:

**Figure supplement 1.** TSLP protein expression during acute LCMV infection.

after infection with an OVA-expressing influenza virus, there were fewer *Crlf2⁻/⁻* OT I cells during the primary infection, indicating that TSLP enhances primary CD8⁺ T-cell responses (*Shane and Klonowski, 2014*). Thus, the reported roles of TSLP in CD8⁺ T-cell responses during primary influenza infection have been somewhat variable, possibly at least in part due to variability in the experimental models/design/animal facilities. Here, we found that TSLP acts directly on CD8⁺ T cells to limit their responses in most tissues during primary influenza infection, with more virus-specific *Crlf2⁻/⁻* cells than virus-specific WT cells. We additionally show that after acute systemic infection with LCMV Armstrong virus, TSLP had a modest effect on primary CD8⁺ T-cell responses, with only a slight increase in *Crlf2⁻/⁻* cells in the spleen at day 8 p.i. Thus, TSLP can limit CD8⁺ T cell responses during primary viral infections, but the effect varied based on the tissue assessed and the type of viral infection. The apparent differences in the various studies of TSLP's impact on CD8⁺ T cells during primary influenza infection may be due at least in part to the tissues sampled (e.g., *Yadava et al., 2013* only assessed responses in BAL fluid). Additionally, the use of influenza viruses with differing pathogenicity in mice might affect the results, as *Shane and Klonowski, 2014* used a strain of influenza, x31, which is less pathogenic than the PR8 strain that we used. Overall, the effect of TSLP on CD8⁺ T cells during the effector phase of a primary acute infection may vary according to the viral infection and tissues analyzed, but our data support the potential for an inhibitory effect by TSLP.

Knowledge of the action of TSLP on memory CD8⁺ T-cell responses has been limited. We found that TSLP does not affect the development/maintenance of memory CD8⁺ T cells after primary influenza infection. In contrast, however, we observed differences after systemic acute LCMV infection, with increased *Crlf2⁻/⁻* virus-specific cells in the blood and lymph nodes at a memory time point post-infection. Importantly, TSLP limited memory CD8⁺ T cell recall responses, with enhanced cellular responses in multiple tissues of *Crlf2⁻/⁻* T cells following either secondary influenza infection or LCMV systemic infection. Interestingly, RNA-Seq data indicated that TSLP suppresses several genes that are related to cell cycle, apoptosis, or protection from virus in influenza infection, with an increased number of virus-specific *Crlf2⁻/⁻* CD8⁺ T cells.

Thus, while TSLP appears to have variable direct effects on primary CD8⁺ T cell responses, possibly depending on the context of infection, here we reveal that TSLP differentially affects naïve and memory cell homeostasis. TSLP enhances naïve CD8⁺ T cell survival in vitro and homeostasis in vivo, but memory CD8⁺ T cell responses are negatively controlled by TSLP, highlighting a key difference between the two cell types. We also found that TSLP uniformly diminished the CD8⁺ T cell responses to secondary acute viral infection in all tissues examined in both pulmonary influenza infection and acute LCMV systemic infection, underscoring a greater effect for TSLP on secondary CD8⁺ T cell responses than primary responses. These findings have potential implications for better controlling secondary responses to viral infection.

## Materials and methods

### Mice

Six to ten week old female C57BL/6 mice were obtained from The Jackson Laboratory. P14 TCR transgenic mice were provided by Dr. Dorian McGavern (NINDS/NIH) and were bred to C57BL/6

Thy1.1$^+$ congenic mice from Jackson Laboratories (B6.PL-Thy1$^a$/CyJ) and with $Crlf2^{-/-}$ mice (*Al-Shami et al., 2004*) in our facility to create congenically-labeled WT and $Crlf2^{-/-}$ P14 mice. All experiments were performed under protocols approved by the National Heart, Lung, and Blood or the National Institute of Neurological Disorders and Stroke Animal Care and Use Committee and followed National Institutes of Health guidelines for the use of animals in intramural research.

## Viruses

Recombinant influenza virus expressing the LCMV gp33-41 epitope (KAVYNFATM) inserted into the NA of A/PR/8/34 (H1N1) (PR8-33) was kindly provided by Dr. Rafi Ahmed (Emory University) (*Mueller et al., 2010*).

## Cell transfer and viral infection

Naïve WT and $Crlf2^{-/-}$ P14 T cells were isolated for use in vitro and in vivo from the spleens of naïve WT and $Crlf2^{-/-}$ P14 mice and purified using a negative selection CD8$^+$ T- cell kit (Stem Cell Technologies). To generate memory CD8$^+$ T cells for use in vivo or in vitro, $5 \times 10^4$ (influenza infections) or $10^5$ (LCMV infections) WT or $Crlf2^{-/-}$ naïve P14 cells were injected i.v. into naïve C57BL/6 mice, and the following day the mice were infected with either $10^3$ EID$_{50}$ of PR8-33 i.n. or $2 \times 10^6$ pfu LCMV Armstrong i.p. At >30 days p.i. the spleens were harvested and memory WT and $Crlf2^{-/-}$ P14 cells purified using a negative selection CD8$^+$ T cell kit (Stem Cell Technologies). A sample of the cells were stained with Thy1.1, Thy1.2, CD8, CD44, V$\alpha$2, and Live/Dead stain to determine the number of P14 T cells. Equal numbers of naïve or memory WT and $Crlf2^{-/-}$ P14 were combined; verification that equal numbers of WT and $Crlf2^{-/-}$ were added was determined by flow cytometry using the same antibodies listed above, and $2.5 \times 10^4$ (influenza) or $10^3$ (LCMV) naive P14 T cells or $10^4$ (influenza) or $10^3$ (LMCV) memory P14 T cells of each population (WT and $Crlf2^{-/-}$) were co-transferred i.v. into naive C57BL/6 mice. On the day following cell transfer, the mice were infected with either $10^3$ EID$_{50}$ of PR8-33 i.n. or $2 \times 10^6$ pfu LCMV Armstrong i.p. For the assessment of the homeostasis of naïve and memory P14 cells in vivo, equal numbers of naïve or memory WT and $Crlf2^{-/-}$ P14 T cells (~1–$2 \times 10^6$ total cells) were co-transferred into naïve C57BL/6 mice in *Figure 2D*.

## Lymphocyte isolation

Lymphocytes were isolated from tissues as previously described (*Masopust et al., 2001*), with some modifications. Briefly, bronchoalveolar lavage (BAL) of the airways was performed with 1 ml of PBS containing 1% BSA prior to perfusion of the lungs with PBS. Lungs were treated with 1 mg/ml Collagenase plus 1 mg/ml DNase (Sigma-Aldrich) in 3 ml of RPMI for 45 min at 37° C. Single cell suspensions were obtained by pushing digested lungs, spleens, and lymph nodes through 40 µM mesh screens (BD Biosciences). Lung lymphocytes were purified by centrifuging (2000 rpm, 4°, for 20 min) on a 44/67% Percoll gradient (Sigma-Aldrich).

## In vitro stimulation assays

Naïve and memory WT and $Crlf2^{-/-}$ P14 T cells were obtained and purified as described above in 'Cell transfers and infections'. $1 \times 10^6$ CD8$^+$ T cells were plated in 0.75 ml/well in 48 well plates or $2 \times 10^6$ CD8$^+$ T cells were plated in 1.5 ml/well total volume in 24 well plates with either medium or 100 ng/ml TSLP (R&D Systems) with or without stimulation. For in vitro stimulation of naïve cells, the plates were pre-coated with 2 µg/ml anti-CD3ε (BioXcell) and 1 µg/ml of soluble anti-CD28 was added. For in vitro stimulation of memory cells, 1 µg/ml of soluble anti-CD3 was added.

## Antibodies and flow cytometry

Single-cell suspensions were stained with anti-mouse Thy1.1-APC, BV421, BV605, FITC, or PE (OX-7), Thy1.2-PerCP, PerCp-Cy5.5 or BV510 (53–2.1), CD8-PE, APC or BV421 (53–6.7), CD44-FITC, PerCP, or BV421 (IM7), KLRG1-BV421 (2F1/KLRG1), V$\alpha$2-PE or APC (B20.1), CD127-APC (A7R34), TNF-$\alpha$- PE-cy7 (MP6-XT22), IFN-$\gamma$- PE-cy7, BV421 or Alexa Fluor 647 (XMG1.2), Annexin V-FITC, and TruStain FcX all were purchased from Biolegend. 7-AAD was purchased from BD Pharmingen. Polyclonal anti-mouse TSLPR-FITC was from R&D Systems and anti-Granzyme B-V450 from BD Horizon (clone GB11). Intracellular staining for granzyme B was performed directly ex vivo or IFN-$\gamma$, TNF-$\alpha$ after a 5 hr in vitro stimulation at 37° with 0.1 µg/ml of gp33 peptide for P14 cells or 0.2 µg/ml of

PR8 NP peptide for non-P14 CD8$^+$ T cells in the presence of GolgiStop and GolgiPlug (BD Biosciences) and cells were fixed and permeablized with Cytofix Cytoperm and Perm wash (BD Biosciences). Cells were analyzed on a LSR II, BD Fortessa or Canto II (BD Immunocytometry Systems), and sorting was done on a BD FACS Aria (BD Immunocytometry Systems). Dead cells were excluded by gated on Live/Dead NEAR IR (Invitrogen).

### IFN-γ and TNF-α protein measurement

Mouse lungs were excised and homogenized using a Minibead beater (Biospec), cleared by centrifugation, and samples were immediately frozen. IFNγ and TNFα protein was determined using the Mouse Inflammation Panel (13-plex) (BioLegend), according to the manufacturer's protocol.

### RNA-Seq analysis and bioinformatics analysis

RNA was isolated from sorted P14 T cells at the indicated time points using the Zymo RNA miniprep kit (Zymo Research), and 500 ng RNA was used for RNA-Seq library preparation with the Kapa mRNA HyperPrep Kit (KK8580, Kapa Biosystems) and indexed with NEXTflex DNA Barcodes-24. After the final amplification, samples were loaded onto 2% E-Gel pre-cast gels (ThermoFisher), and 250 to 400 bp DNA fragments were excised and purified with Zymoclean Gel DNA Recovery Kit (Zymo Research). After quantification by Qubit (Invitrogen), barcoded samples were mixed and sequenced on an Illumina HiSeq 3000 system. Sequenced reads (50 bp, single-end or paired-end) were obtained with the Illumina CASAVA pipeline and mapped to the mouse genome mm10 (GRCm38, Dec. 2011) using Bowtie 2.2.6 and Tophat 2.2.1. Raw counts that fell on exons of each gene were calculated and normalized by RPKM (Reads Per Kilobase per Million mapped reads). Differentially expressed genes were identified with the R Bioconductor package 'edgeR', and expression heat maps were generated with the R package 'pheatmap'.

### Statistics

Data are presented as mean ± SEM. Two-tailed paired students t-test was used for statistical analysis. All statistical analyses were performed using Prism v7 (GraphPad Software, La Jolla, CA). Differences were considered significant when $p < 0.05$.

### Source data

Source files for RNA-Seq in *Figures 3* and *4* are in *Supplementary files 1–4*. The RNA-seq data are available at Gene Expression Omnibus (GEO) under accession code GSE 156875.

## Acknowledgements

This work was supported by the Division of Intramural Research, National Heart, Lung, and Blood Institute and the Division of Intramural Research, National Institute of Neurological Disorders and Stroke. Next generation sequencing was performed in the NHLBI Sequencing core. RE-S. was supported by a Public Interest Incorporated Foundation Scholarship from the MSD Life Science Foundation. We thank Dr. Rosa Nguyen for assisting with harvesting mouse tissues. We thank Dr. Rafi Ahmed, Emory University for valuable suggestions/critical comments.

## Additional information

### Funding

| Funder | Grant reference number | Author |
| --- | --- | --- |
| National Heart, Lung, and Blood Institute | Division of Intramural Research | Warren J Leonard |
| National Institute of Neurological Disorders and Stroke | Division of Intramural Research | Dorian B McGavern |

The funders had no role in study design, data collection and interpretation, or the decision to submit the work for publication.

## Author contributions
Risa Ebina-Shibuya, Formal analysis, Investigation, Writing - original draft, Writing - review and editing; Erin E West, Conceptualization, Formal analysis, Investigation, Writing - original draft; Rosanne Spolski, Formal analysis, Investigation, Writing - review and editing; Peng Li, Formal analysis, Writing - original draft, Writing - review and editing; Jangsuk Oh, Phillip Swanson, Ning Du, Investigation; Majid Kazemian, Daniel Gromer, Formal analysis, Writing - review and editing; Dorian B McGavern, Supervision, Writing - review and editing; Warren J Leonard, Supervision, Funding acquisition, Writing - review and editing

## Author ORCIDs
Risa Ebina-Shibuya (iD) https://orcid.org/0000-0001-9031-5070
Majid Kazemian (iD) https://orcid.org/0000-0001-7080-8820
Warren J Leonard (iD) https://orcid.org/0000-0002-5740-7448

## Ethics
Animal experimentation: All experiments were performed under protocols approved by the National Heart, Lung, and Blood Institute Animal Care and Use Committee (Protocol H-0087R4) or the National Institute of Neurological Disorders and Stroke Animal Care and Use Committee (Protocol 1295-17) and followed National Institutes of Health guidelines for the use of animals in intramural research.

## Decision letter and Author response
Decision letter https://doi.org/10.7554/eLife.61912.sa1
Author response https://doi.org/10.7554/eLife.61912.sa2

# Additional files
## Supplementary files
• Supplementary file 1. Differentially expressed genes from sorted naïve and memory CD8$^+$ T cells 4 or 24 hr after incubation with medium or TSLP, without TCR stimulation. Related to *Figure 3A*.

• Supplementary file 2. Differentially expressed genes from sorted naïve and memory CD8$^+$ T cells 4 or 24 hr after incubation with medium or TSLP, with TCR stimulation. Related to *Figure 3B*.

• Supplementary file 3. RNA-Seq comparison between naïve and memory CD8$^+$ T cells 24 hr after incubation with TSLP, without TCR stimulation. Related to *Figure 3C–E*.

• Supplementary file 4. Differentially expressed genes of WT versus *Crlf2*$^{-/-}$ mice, from CD8$^+$ T cell recall responses at day eight after secondary infection. Related to *Figure 4E–G*.

• Transparent reporting form

## Data availability
Sequencing data are available in the Supplementary files 1–4 and have been deposited in GEO, accession codes GSE156875.

The following dataset was generated:

| Author(s) | Year | Dataset title | Dataset URL | Database and Identifier |
|---|---|---|---|---|
| Ebina-Shibuya R, West EE, Spolski R, Li P, Oh J, Kazemian M, Gromer D, Swanson P, Du N, McGavern DB, Leonard WJ | 2020 | Thymic stromal lymphopoietin limits primary and recall CD8+ T-cell anti-viral responses | https://www.ncbi.nlm.nih.gov/geo/query/acc.cgi?acc=GSE156875 | NCBI Gene Expression Omnibus, GSE156875 |

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
