## [Decision Letter]

**Acceptance summary:**

The role of the cytokine TSLP on CD4^+^ T cells responses has been extensively studied, but less is known about its' role in CD8^+^ responses. This paper shows that CD8^+^ T cells lacking the TSLP receptor are impaired in their response to viral infection, particularly during secondary challenge responses. They also show that naïve and memory T cells show distinct transcriptional responses to TSLP, providing insight into the function of this important cytokine in CD8^+^ T-cell response.

**Decision letter after peer review:**

Thank you for sending your article entitled "Thymic stromal lymphopoietin limits primary and recall CD8^+^ T cell anti-viral responses" for peer review at *eLife*. Your article is being evaluated by three peer reviewers, one of whom is a member of our Board of Reviewing Editors, and the evaluation is being overseen by Tadatsugu Taniguchi as the Senior Editor.

As you and your coauthors noted, there is a notable discrepancy between your study and the work of Shane et al., which demonstrated that TslprKO CD8 T cells were reduced at memory timepoints in the broncheoalveolar lavage and lungs using the OT-I TCR transgenic and OVA-expressing influenza virus. While the authors suggest that differences in models could explain the discrepancy between the two studies, it is surprising that these groups arrived at diametrically opposed results using the influenza model (albeit different strains). While the current results with TslprKO mice are strengthened by using the LCMV model, there is concern that there may be unexpected differences between TslprKO mouse lines used by the two groups (Ziegler lab and Leonard lab lines). Of particular concern is the possibility that passenger mutations in one of these lines could have created a confounding variable. For example, see PMCIDs PMC4364188 [Dock8 mutation in Nlrp10KO mice, Eisenbarth group], PMC3326475 [Dock2 mutation in Irf5KO mice, Bhattacharya group], and PMC3660499 [Sox13 mutation in B6.SJL congenic mice, Cyster group]. Both Shane et al. and Ebina-Shibuya, West, et al. appear to have maintained independent crosses of WT and TslprKO mice to P14, which could in theory allow a confounding mutation to be maintained in the TslprKO P14 line. Because of the relatively subtle nature of the observed numerical changes, it is also possible that the observed differences between congenically distinct donor cells arise not from TSLPR, but from background genetic differences between Thy1.1/Thy1.1 and Thy1.1/Thy1.2, which could have been addressed by using co-transfer of congenically distinct WT P14 T cells to a separate group of mice as a control. The best way to address all of these concerns would be to block TSLP signaling using the M702 antibody (see Kuan and Ziegler, Nat Immunol, 2018). This would provide independent validation of the effects of TSLP signaling loss, with the additional benefit of avoiding changes in CD8^+^ T cell development. Could the authors either singly transfer WT P14 or cotransfer WT and TslprKO P14 to recipients and treat with control Ig or anti-TSLP antibodies as an independent means to validate their findings? In the case of single transfer, donor WT P14s would be expected to increase in mice treated with anti-TSLP abs. In the case of cotransfer, donor WT P14s would be expected to expand equally to TslprKO P14s in mice treated with anti-TSLP abs. If the authors had Thy1.1^+^ Thy1.2- TslprKO P14 mice available as a separate line, this would be another means to satisfy these points (study currently uses Thy1.1^+^ Thy1.2^+^ TslprKO P14 mice). I realize that such experiments are challenging in our current environment, but the reviewers consider addressing these issues critical for publication.

---

## [Author Response]

As you and your coauthors noted, there is a notable discrepancy between your study and the work of Shane et al., which demonstrated that TslprKO CD8 T cells were reduced at memory timepoints in the broncheoalveolar lavage and lungs using the OT-I TCR transgenic and OVA-expressing influenza virus. While the authors suggest that differences in models could explain the discrepancy between the two studies, it is surprising that these groups arrived at diametrically opposed results using the influenza model (albeit different strains). While the current results with TslprKO mice are strengthened by using the LCMV model, there is concern that there may be unexpected differences between TslprKO mouse lines used by the two groups (Ziegler lab and Leonard lab lines). Of particular concern is the possibility that passenger mutations in one of these lines could have created a confounding variable. For example, see PMCIDs PMC4364188 [Dock8 mutation in Nlrp10KO mice, Eisenbarth group], PMC3326475 [Dock2 mutation in Irf5KO mice, Bhattacharya group], and PMC3660499 [Sox13 mutation in B6.SJL congenic mice, Cyster group]. Both Shane et al. and Ebina-Shibuya, West, et al. appear to have maintained independent crosses of WT and TslprKO mice to P14, which could in theory allow a confounding mutation to be maintained in the TslprKO P14 line.

We agree that it is possible that a passenger or other mutation could be influencing the results in one or both of the studies, although both of the *Tslpr* KO mouse strains have been shown to behave similarly, for example related to Ova-induced allergic inflammation (PMID 16172260 and 16142237). Although the basis for the differences remains unclear, it is possible, as we suggested, that differences in the model and the strains of influenza could affect the outcome, and moreover the animals were in animal facilities at different institutions, so that differences in the microbiome might be another contributing factor as well.

Because of the relatively subtle nature of the observed numerical changes, it is also possible that the observed differences between congenically distinct donor cells arise not from TSLPR, but from background genetic differences between Thy1.1/Thy1.1 and Thy1.1/Thy1.2, which could have been addressed by using cotransfer of congenically distinct WT P14 T cells to a separate group of mice as a control. The best way to address all of these concerns would be to block TSLP signaling using the M702 antibody (see Kuan and Ziegler, Nat Immunol, 2018). This would provide independent validation of the effects of TSLP signaling loss, with the additional benefit of avoiding changes in CD8^+^ T cell development. Could the authors either singly transfer WT P14 or cotransfer WT and TslprKO P14 to recipients and treat with control Ig or anti-TSLP antibodies as an independent means to validate their findings? In the case of single transfer, donor WT P14s would be expected to increase in mice treated with anti-TSLP abs. In the case of cotransfer, donor WT P14s would be expected to expand equally to TslprKO P14s in mice treated with anti-TSLP abs. If the authors had Thy1.1^+^ Thy1.2- TslprKO P14 mice available as a separate line, this would be another means to satisfy these points (study currently uses Thy1.1^+^ Thy1.2^+^ TslprKO P14 mice). I realize that such experiments are challenging in our current environment, but the reviewers consider addressing these issues critical for publication.

Two different approaches were suggested: using a blocking antibody (M702) to TSLP or using Thy1.1^+^ Thy1.2^-^*Tslpr* KO P14 mice as a second line. Regarding the first approach, we have concerns that the experiment could be technically challenging, as it would require multiple injections of an antibody, where one would also have to assure that a sufficient level of blocking activity was achieved. Moreover, an experiment using a TSLP blocking antibody would block the actions of TSLP on all cell types and thus would not allow one to infer the direct actions of TSLP on CD8^+^ T cells. Accordingly, we prefer the second approach, and in fact, we already had performed the requested experiment. We apologize for having erred in our labeling of Figure 1D and the cartoon schematic in Figure 1B; although all but one of the included experiments were as indicated, one experiment was in fact included that used Thy1.1^+^ Thy1.2^-^*Tslpr* KO P14 mice. In the appended figure, the data in Figure 1D are now divided to show the data where recipients were Thy1.1^+^ Thy1.2^+^*Tslpr* KO P14 mice in panel A and the experiment in which recipients were Thy1.1^+^ Thy1.2^-^*Tslpr* KO P14 mice in panel B. As you can see, the trend is the same, providing confirmation of the validity of our findings as requested. Figure 1D represents the composite of these data.